# The implementation of the Workforce Indicators of Staffing Needs (WISN) method to improve access to health workforce in selected South-East Asian countries

Sikhumbuzo A. Mabunda[1,2,3,4]*, Mona Gupta[5], Roshan Sampath[6], Rohina Joshi[1,2,7]

1 School of Population Health, University of New South Wales, Sydney, Australia, 2 Department of Public Health, Walter Sisulu University, Mthatha, South Africa, 3 The George Institute for Global Health, University of New South Wales, Sydney, Australia, 4 Global Centre for Human Resources for Health Intelligence, Walter Sisulu University, Mthatha, South Africa, 5 National Health Systems Resource Centre, Ministry of Health and Family Welfare, India, 6 International Organization for Migration, Colombo, Sri Lanka, 7 The George Institute for Global Health India, New Delhi, India

* s.mabunda@unsw.edu.au

## Abstract

The Workload Indicators of Staffing Needs (WISN) is a tool used to estimate health workforce requirements for national or sub-national regions and health facilities. We determined the implementation of WISN for six countries in the World Health Organization South-East Asia Region (WHO-SEAR). This research ascertained whether WISN's recommendations were adopted and operationalised after the initial assessment. This will then help policymakers identify implementation barriers and enablers for the successful implementation of WISN for the improvement of access to health workforce. We used a multi-method approach comprising qualitative, in-depth, semi-structured interviews, literature reviews and document reviews. First, we conducted a desktop review to understand the context of WISN implementation in Bangladesh, Bhutan, India, Indonesia, Nepal and Sri Lanka. Second, we interviewed a key policymaker responsible for the implementation of WISN in Sri Lanka, Bangladesh, Bhutan and India. Interviews were undertaken virtually, in English using Microsoft Teams virtual software and auto transcribed and recorded using Microsoft Teams. Literature reviews were conducted using electronic databases, documents and reports were sourced from the WHO-SEAR office and/or country of interest's focal persons. Findings from the different methods were synthesized, triangulated and presented using four themes, namely, initial implementation, key findings, primary lessons and directions forward. This study found a high utility of WISN on informing policymakers of the health workforce needs or surplus once the service activity standards were determined. However, all the countries did not have pre-defined service standards and they did not always have the required service data or health information. For WISN to be of utility, countries need to first strengthen

**Data availability statement:** Data used in this publication are incorporated within the manuscript.

**Funding:** This work was supported by the WHO South East Asia Regional Office (to SAM and to RJ). The funders had no role in study design, data collection and analysis, decision to publish, or preparation of the manuscript.

**Competing interests:** I have read the journal's policy and the authors of this manuscript have the following competing interests: MG is an employee of the Indian Ministry of Health. All other authors declare no conflict of interest.

their health workforce information systems and digitize workload standards. Furthermore, the government treasury and the employing ministry must be involved at early stages of the planning process to ensure easy phasing in and adaptation of recommendations.

## Introduction

Health workforce is the backbone to the implementation of primary health care and is the foundation for universal health coverage (UHC) [1]. Therefore, it is essential to plan the number and the skills mix required to achieve the country's national health goals [2,3]. Lack of health workforce planning may lead to shortage or surplus of workforce, maldistribution of workforce resulting in poor health outcomes. Evidence-based health workforce planning is fundamental in guiding the actions of policymakers towards achieving UHC [4]. Considering the need for efficient and effective health workforce management, the World Health Organization (WHO) developed the Workload Indicators of Staffing Needs (WISN), a method to calculate human resource requirements for health systems [5]. Unlike traditional techniques such as the population ratio methods, WISN considers the healthcare delivery models and population demands [5]. It provides two indicators to assess staffing needs: (a) the gap between current and required number of staff and (b) WISN ratio, a measure of workload pressure on health workers [6]. Staffing requirements are calculated by the workload at the facility and the standard workload for a cadre. It helps estimate the number and cadre of health workers required for health services at a facility and estimates the pressure of workload on health workers. This method has been applied in several countries at the national and sub-national levels [7–11].

The use of WISN has been welcomed as a health workforce planning tool since it considers population health needs, unlike traditional strategies that calculate population health needs based only on the size of the population [12,13]. These strategies assume that populations of the same size will have similar health needs and fails to adequately plan for suitable skills mix that will serve the prevailing or projected population's burden of diseases [12,13]. Furthermore, the utility of WISN is not only limited to the determination of skilled health professionals (e.g., physicians, nurses and midwives) but it can be extended to be used in the determination of public health cadres (e.g., community health workers, volunteers, etc.) and midlevel workers who are the backbone of primary health care provision in many LMICs [5,14–18]. Notwithstanding WISN's advantages, the process is labour intensive, the tool requires quality data with supportive health information systems, relies on the cooperation and coordination of multiple stakeholders, needs subjective skills to develop informed service standards, independent performance monitoring tools and separate supportive or complimentary development, recruitment and retention strategies [13]. Furthermore, whilst WISN will determine the required health workforce and skills mix, it will not cost this workforce, and it will also not propose a strategy on how the suggested health workforce can be phased in to ensure practical implementation

[13]. This latter example is one of the biggest criticisms of the WISN strategy, for instance, South Africa's ministry of health previously determined the health workforce needs in primary care but could not use those recommendations due to lack of funds [13].

After sub-Saharan Africa, South-East Asia has some of the most acute health workforce shortages in the world [19]. This is also a region where demographic and epidemiological changes are occurring at a fast pace and the need for a trained and available health workforce is crucial. For instance, even though these countries are LMICs, noncommunicable diseases (NCDs) caused nearly two thirds of all deaths in WHO South-East Asian countries [20]. It is also important noting that this region is home to over a quarter of the world's population, which shares a disproportionate burden of disease and mortality [21]. It is projected that this region will have 40% of the global health workforce shortages that are estimated for 2030 [22]. These shortages are mostly attributed to maldistribution, emigration of the health workforce, substandard working conditions, poor planning, training challenges, gaps in health workforce data, and weak health workforce governance [22].

Some of the southeast Asian region (SEAR) countries have used WISN to assess adequacy of their health workforce [2,11,23–25]. The experience with the use of WISN has been varied [17] and not all these countries implemented WISN beyond the initial needs assessment [1,6–9]. Therefore, this study aimed to ascertain the implementation of the Workload Indicators of Staffing Needs (WISN) recommendations for primary health care health workforce planning for countries in the World Health Organization South East Asia Region (WHO-SEAR). The countries of interest are all classified as low- and middle-income countries (LMICs) with high health workforce needs, thus requiring efficient health workforce planning strategies. There are literature gaps on the use of WISN recommendations beyond the initial determination of the human resources for health gaps as was the case in South Africa [17]. This study will therefore help to determine whether WISN's recommendations were adopted after the initial assessments in countries in the WHO South-East Asia region. It is then hoped that these findings will help policymakers identify implementation barriers and enablers for the successful implementation of WISN for the improvement of access to health workforce in the countries studied, those who have used WISN.

## Materials and methods

### Ethics statement

The study was reviewed and approved by the Human Research Ethics Committee at the University of New South Wales, Faculty of Medicine and Health in Australia. All participants provided written informed consents.

### Study design

We used a multi-methods approach [26–31] comprising of a literature review, document reviews of WISN reports and semi-structured, in-depth interviews to understand the implementation of WISN in Bangladesh, Bhutan, India, Indonesia, Nepal and Sri Lanka. All the reviews had no language or date restrictions. We conducted a desktop review of each country and its health indicators to understand the context of each country. Then, we reviewed the WISN reports from each country including the planning, sampling, results, recommendations, and implementation of recommendations. Finally, we conducted in-depth interviews of policymakers between 17 June 2022 and 30 August 2022 to understand the implementation of WISN in the respective countries (See S1 Text for the interview guide). This strategy was chosen as it was the most practical, efficient and comprehensive strategy to ascertain the implementation of WISN recommendations in these six South East Asian countries.

Literature reviews were conducted using electronic databases (Medline, CINAHL, Google Scholar and Google), documents and reports were sourced from the WHO-SEAR office and/or country of interest's focal persons who were either employed by the Ministry of Health or WHO. This review strategy was used to ensure that all relevant studies (whether published or not) would be included.

The research team contacted a key policymaker responsible for the implementation of WISN in Sri Lanka, Bangladesh, Bhutan and India. All participants were provided with the participant information statement and consent form via email. Participants were given sufficient time to consider their participation and were advised to contact the researcher(s) if they had any questions. They were asked to email the signed electronic copy of the consent to the researcher(s) prior to data collection. Participants were consented for audio-recording of the interviews. All interviews took place virtually via Microsoft Teams and the same software was used for transcription.

## Sampling and, identification of literature and reports

The World Health Organization South East Asia Regional Office (WHO-SEARO) helped with the identification of policymakers and stakeholders who had led the implementation of WISN in Bangladesh, Bhutan, India, Indonesia and Sri Lanka. WHO-SEARO further shared country reports on the introduction and implementation of WISN in these countries. Medline, CINAHL, Google and Google Scholar searches were conducted to identify all literature on WISN from these six WHO South East Asian countries.

## Data collection

The literature and report reviews were extracted using a Microsoft Excel spreadsheet that collected information on the nature of the publication, the year of publication, objectives of the report or publication, methods used, processes used to set up working groups, the WISN ratio, the actual staff numbers, the required staff numbers, actions taken after WISN implementation, and the report or publication's conclusions. Five interviews from four countries were conducted virtually by RJ using Microsoft Teams and lasted an average of 47 minutes. The research team did not manage to get into contact with an Indonesian and Nepalese focal people for WISN using multiple strategies. Two participants were interviewed separately for Bhutan. Participants were asked to share their involvement with the WISN implementation in their respective country, the planning and consultation processes involved, the process of determination of packages of care, the processes involved in the development of service standards and skills mix, all steps involved in the implementation of WISN including the sampling/selection of States/Provinces and/or districts, successes derived from the process, challenges experienced with the implementation of WISN, and an overview of how the WISN recommendations have been used.

## Data analyses

A narrative approach was used for the review of reports and literature based on comparison of the methods used, sampling strategies, implementation of tool, successes, challenges, areas of improvement and overall results. Qualitative data were transcribed verbatim and analysed thematically. Two researchers reviewed all transcripts and analysed the interviews manually using an inductive, thematic content analysis approach. The analysis followed a six-step approach of familiarisation, coding, theme development, review of themes, defining themes and reporting. The overall themes from the interviews were shared with the participants to ensure that their perspectives were incorporated correctly. The results synthesised and triangulated the findings of the reports and interviews and presented them by country as there were different variables and approaches in each country. Three themes are used for the presentation of results, namely, Overview and Initial implementation, Key findings, Primary lessons and directions forward. These themes focus on the process flow of WISN implementation.

## Role of the funding source

This work was funded by WHO SEARO. Data collection, analysis and interpretation was led by researchers in consultation with country-based researchers and policymakers.

## Results

All the six countries are low- middle-income countries with health workforce shortages (Table 1). All the countries have adopted the United Nation's Sustainable Development Goals and aim for UHC through decentralised health systems (except for Bangladesh which has centralised governance). The health system in all the countries is serviced by the public and private sectors and includes both western and traditional medicine.

Five policymakers and stakeholders from Bangladesh (1), Bhutan (2), India (1) and Sri Lanka (1) of whom two were female were interviewed. The literature search yielded seven peer reviewed articles from three countries (Bangladesh, India and Sri Lanka). Countries with peer reviewed articles on WISN were Bangladesh (three with a correction for one) [25,32–34], India (two) [11,35]and Sri Lanka (one) [23]. A published WHO report on the WISN implementation of Indonesia was also found through electronic database literature search [36]. In addition, other technical reports were acquired from Bangladesh (two), India (one), Nepal (one) and Sri Lanka (two) [24,37–41].

All the six countries used WISN with technical support from the WHO. Bangladesh had the most experience in using WISN while India's government had successfully piloted WISN for the first time. The findings from the six countries varied by context, however, all countries indicated significant gaps in their health workforce.

The main challenge faced was the lack of documentation of workload standards and the lack of standardised data for activities conducted by staff. There were vast variations in time required to undertake activities (which is not documented), these vary by experience, context and infrastructure available. Moreover, there are no standards for tasks which are shared between team members, especially at the primary care level and there is insufficient consideration of the unique circumstances and needs in rural areas. All the studies have made recommendations to the respective Ministries to increase the staff numbers, of which, India has approved the budget for additional staff. The Bangladesh Ministry of Health has requested the Finance Ministry for additional funding, while Bhutan has not made any changes. Although the ministries in Sri Lanka, Nepal, Indonesia and Bangladesh have accepted the recommendations, there are no changes implemented yet. Details of the findings are provided in Table 2.

### Bangladesh

**Overview and initial implementation.** When including a correction of an article, six WISN articles and reports were reviewed for Bangladesh [24,25,32–34,39]. The Bangladesh Health workforce strategy 2015 recommends determining health workforce needs based on workload staffing approach [24,33]. The one study discussed with a respondent is one where they aimed to assess the current staffing needs of the public sector health facilities in two districts: Mymensingh and Barishal. WISN was planned under a Steering Committee, Technical taskforce, and Expert Working groups. All

**Table 1. Overview of country context.**

| Country | Bangladesh | Bhutan | India | Indonesia | Nepal | Sri Lanka |
|---|---|---|---|---|---|---|
| Population, 2023 | 173·0 million | 787,424 | 1·4 billion | 277·5 million | 30·9 million | 21·9 million |
| GDP; USD, 2023 | 446·4 billion | 3·0 billion | 3·4 trillion | 1·1 trillion | 41·2 billion | 74·1 billion |
| Human development index, 2022 | 0·67 | 0·68 | 0·64 | 0·71 | 0·60 | 0·78 |
| Population density per km², 2023 | 1,165 | 21 | 435 | 146 | 210 | 334 |
| Physician density (per 1000 population), 2021 | 0·7 | 0·6 | 0·7* | 0·7 | 0·9 | 1·2 |
| Nursing and midwifery density (per 1000 population), 2021 | 0·6 | 2·2 | 1·7* | 4·0* | 3·5 | 2·4 |
| Life expectancy (years), 2023 | 74·0 | 72·5 | 72·0 | 71·1 | 71·7 | 76·8 |
| Under-5 mortality rate, 2023 | 24·1 | 23·5 | 30·5 | 21·3 | 27·7 | 6·3 |

*2020 data

**Table 2. Overview of results from the five countries.**

| Country | Facilities involved | WISN results | Use of recommendations |
|---|---|---|---|
| Bangladesh | Primary, secondary and tertiary level, two districts | Health workers across all three levels are working under extremely high workload pressure | • Recommendations for additional health workers have been endorsed by the Ministry of Health. Funding is yet to be sanctioned.<br>• Staffing norms/standards of different health facilities were updated in 2021. The revised staffing standards is awaiting Ministry of Public Administration approval.<br>• Based on the findings, the Ministry of Health and Family Welfare drafted health service activity standards for sub-district level health workforce in 2022. This reference book is now at finalisation stage. |
| Bhutan | Primary, secondary and tertiary level | Adequate: dentists, surgeons, paediatricians and radiology technologists.<br>Surplus: nurses, dental technicians and OT staff and physiotherapists.<br>Shortage: lab staff, gynaecologists, medical officers, health assistants. | • No changes have been made based on the recommendations.<br>• Recommendations from WISN have not been implemented. |
| India | Secondary and tertiary level, 2 states | Adequate: gynecologists, Surplus: paediatricians<br>Shortage: nurses, lab technicians | • Funding allocated for the understaffed positions. No action taken for surplus posts.<br>• Follow-up not been completed to evaluate if the recommendations are implemented. |
| Indonesia | Hospitals, 4 provinces | Adequate: not reported<br>Surplus: not reported<br>Shortage: nurses, general practitioners, lab technicians and lung specialists | • No information about use of the recommendations |
| Nepal | Primary health care, 3 provinces | Adequate: staff nurses, auxiliary heath workers<br>Surplus: auxiliary nurse midwives<br>Shortage: health assistants | • Recommendation to reallocate/ transfer the surplus staff.<br>• No information about use of the recommendations |
| Sri Lanka | Primary health care, 25 - Districts | WISN ratio was not calculated for PHC workforce except midwives.<br>Shortage: midwives | • No information about use of the recommendations |

three levels of the health system in the two districts were assessed: tertiary, secondary and primary health centres (15 in Barishal and 14 in Mymensingh).

**Key findings.** The health workforce in Bangladesh was found to work under extremely high workload pressure across primary, secondary and tertiary levels (Table 3). The highest workforce required were nurses in the obstetrics and gynaecology (O&G) department with a WISN difference of -73·66.

**Primary lessons and directions forward.** WISN recommended recruitment of health workforce to fill the vacant positions and the need for the health workforce to be relieved of administrative and support activities so that they can focus on clinical care. It recommended reallocation of staff from low to high workload health facilities and where facilities faced extreme staff shortage, e.g., Obstetrics and Gynaecology department. A long-term development and recruitment policy response was recommended to fulfil the needs due to the short supply of Nurses and Medical Technologists. The Bangladesh Ministry of Health supports the staffing needs report and have accepted all recommendations as they are backed by evidence and the report wants to improve the quality of healthcare [24,25,32–34,39]. The report awaits support from the Finance Ministry.

**Bhutan**

**Overview and initial implementation.** According to two interview respondents, Bhutan used WISN to determine staffing levels and identify gaps in the selected facilities across primary, secondary and tertiary care. This began with a

**Table 3. Workload indicator staffing needs ratio and health workforce requirements for Bangladesh (2019).**

| Workload (WISN Ratio) | Barishal Medical College | Mymensingh Medical College | Barishal District | | Mymensingh District | |
|---|---|---|---|---|---|---|
| | | | Wazirpur Upazila | Bakergunj Upazila | Nandail Upazila | Trishal Upazila |
| Extremely high (0·10-0·29) | 28 | 24 | 20 | 20 | 6 | 11 |
| Very high (0·30 and 0·49) | 19 | 18 | 10 | 25 | 6 | 26 |
| High (0·50 and 0·69) | 13 | 18 | 30 | 10 | 29 | 26 |
| Moderately high (0·70 and 0·89) | 16 | 21 | 20 | 20 | 29 | 4 |
| Normal (0·90-1·19) | 14 | 18 | 5 | 10 | 0 | 22 |
| Low (≥1·20) | 11 | 3 | 15 | 15 | 29 | 11 |

nine (9) facility WISN pilot study by purposively sampling a fraction of cadres between March and August 2019. This was under the leadership of a Steering Committee which was supported by a Technical Taskforce and an Expert Working Group. Methods included literature reviews, key informant interviews and a time and motion study to gather sufficient background information about working standards.

**Key findings.** The study identified mixed results of adequacies, shortages and surpluses that varied across health facilities and categories. Dental surgeons, paediatricians and the radiology technologists had adequate staff requirements. The highest shortages were recorded by the laboratory staff (-21), gynaecologists (-6), General Duty Medical Officers (-5) and health assistants (-4). Surpluses were identified for nurses (+62), dental technicians (+22), operating theatre technicians (+17) and physiotherapists (+13).

**Primary lessons and directions forward.** This process derived short, medium and long-term recommendations as summarised in Table 4.

Even though no direct changes have been made based on the recommendations, the findings were referred to during the human resources for health (HRH) strategy discussions, service standards and HRH standards review and development.

### India

**Overview and initial implementation.** According to a report and a policymaker interviewed, the aim of using WISN in India was to pilot it in two states (Kerala and Meghalaya) and one union territory (Chandigarh) [38]. The planning and implementation were led by the National Health Systems Resource Centre in partnership with the respective states [38]. One district hospital from each of the three States (Ri-Bhoi, Meghalaya, Pathanamthitta, Kerala and Chandigarh, Chandigarh) was included [38]. As the study was undertaken during COVID-19, Meghalaya had to be dropped due to the COVID-19 pandemic related logistical challenges.

India tried to pilot WISN earlier using the available tools and resources, but the study could not be concluded successfully. The key challenge faced in piloting was lack of technical knowledge to implement WISN [17]. Using the lessons learnt from that experience, this time, technical assistance was sought from WHO to undertake WISN in three districts. The implementation process involved having a) a steering committee, b) a technical taskforce and c) an expert working group. Roles and responsibilities of these committees were clearly defined. While collaborative, the process adopted a top-down approach which involved discussions with all the major stakeholders, both at the national and state level. Training was conducted by international experts supported by WHO-SEARO and WHO India. The study was conducted at a cadre level across two district hospitals.

**Key findings.** Overall, the WISN study showed adequate number of gynaecologists, oversupply of paediatricians and under supply of nurses and lab technicians (Fig 1). Poor documentation and poor reporting of workload and activities may have impacted the results gathered from WISN. The service standards created for gynaecologists may be inflated as the

**Table 4. Short, medium and long-term WISN recommendations in Bhutan.**

| Short-term | Medium-term | Long-term |
|---|---|---|
| • Conduct a larger representative study.<br>• Cascade the WISN training to the district and facility levels to empower facility managers.<br>• Redeploy health workers from health facilities with surpluses and extremely low workloads to those with shortages and high workload pressures.<br>• Standardise health services provided in health centres of similar levels of care to enhance equity.<br>• Capture all services provided by all health cadres.<br>• Strengthen health service processes and information systems.<br>• Ensure equitable training opportunities for staff.<br>• Strengthen primary care services to ensure efficiency of the health system. | • Review job descriptions for the subcategories of staff whose roles overly overlap.<br>• Consider task shifting/sharing to alleviate shortages in the short-term as the country works towards increasing production of the needed cadres.<br>• Consider revising some of the curricula of the technicians trained in the country to address context-based issues to accommodate the country's needs especially primary care services.<br>• Use the WISN results to revise staffing norms and standards to correspond to the health needs and to plan for future staffing.<br>• Develop a comprehensive health information system that is interoperable, integrated and interlinked at all levels of care to support health planning. | • Consider merging some cadres who are doing the same tasks.<br>• Review some of the human resources for health practices and service delivery models through policies and plans to implement people-centred comprehensive quality services.<br>• Reconsider the roles of the village health workers in primary care with definite scope of work for ease of quantifying their contribution to the health system.<br>• Strengthen the production capacity of the country to increase the number of mid-level cadres, while also increasing the number of required specialists based on the country's current and future needs. |

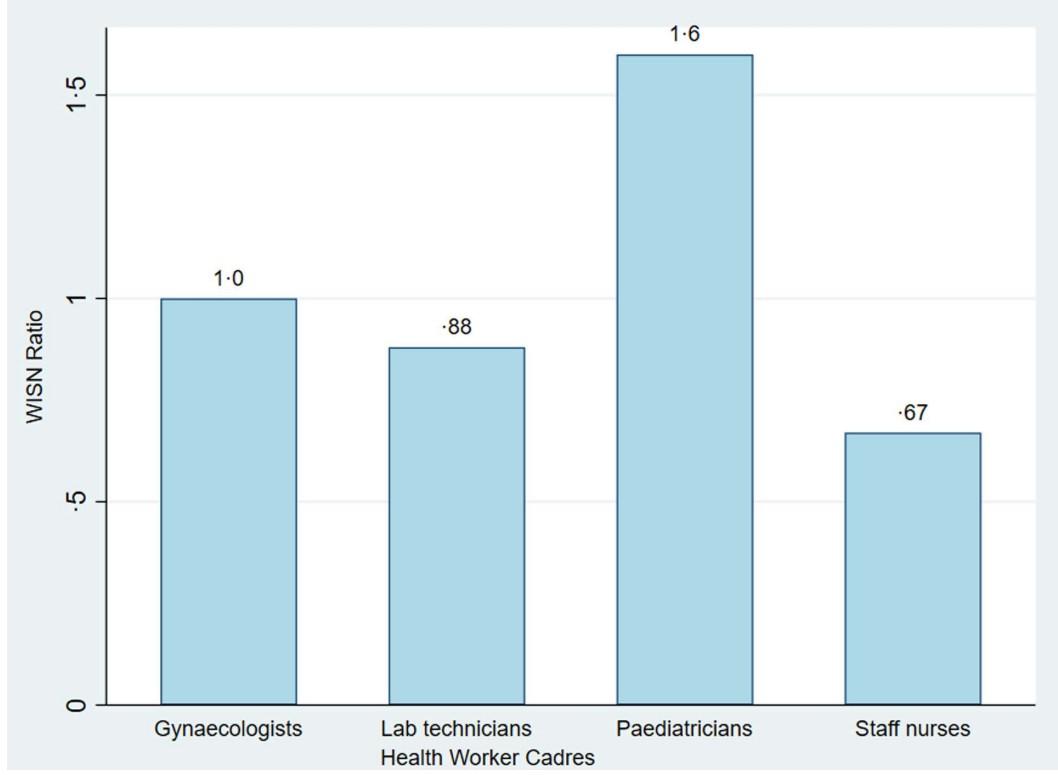

**Fig 1. WISN ratio results for four cadres in two Indian health facilities.**

cadres assisting gynaecologists were not assessed. Equally, the time taken for procedures listed by the paediatricians may be less than the real number due to poor reporting and documentation mechanisms. Consequently, the WISN report should be reviewed understanding the context of the health facility.

**Primary lessons and directions forward.**  The Indian Public Health Standards (IPHS) [42] provide a reference point for healthcare infrastructure planning in public health facilities. While the IPHS were formulated based on expert opinion, rather than evidence base, the numbers recommended by IPHS are believed to be aligned with the findings of WISN. According to a policymaker, findings from one facility in a state cannot be generalised to other parts of the state or the nation as it does not take into consideration the diversity and a range of factors including competence, infrastructure and leadership of health services. Even though all the materials for WISN are freely available online, it was not easy to implement without technical assistance from the WHO. Data capture is challenging as the health workforce is not trained and does not have the time to record activities. Furthermore, there was no hospital management information system available at the time. This is especially difficult when dealing with task-sharing across different team members of departments or facilities. As all medical teams work together, there is often some overlap of duties which were not captured by WISN. WISN assumes that service standards and the number of hours for each activity are available and standardised for different cadres. This is not the case in India. The time needed to perform a task is based on the competency of the doctors, therefore, when setting these standard times, one needs to ensure that the committee making the decision on service standards for medical doctors is composed of medical doctors who have practiced in similar settings and understand different levels of competencies. The use of WISN is limited when facilities are over-staffed as the oversupply is not redistributed to areas of need. In fact, the argument is made that due to poor recording of activities, the inputs are incorrect thereby giving a false sense of a facility being over-staffed. Furthermore, WISN considers the average time taken for activities, however, it doesn't take into consideration the ethos of patient centric services. While allocating the time required for each activity, the focus is on the health worker, and not how much time a patient may need for the consultation.

Results were communicated with the two districts and more funds have been allocated for the posts which are under-staffed. No action is taken for the posts which had an oversupply as the health facility team had argued that the actual case load was higher and better documentation of cases would capture accurate numbers.

## Indonesia

**Overview and initial implementation.**  Indonesia's Ministry of Health conducted WISN during COVID-19 to understand if health facilities provided a safe environment to promote safe patient care related to COVID-19 and, protect the health and wellbeing of the health workforce; and to quantify the required health workforce based on the pressures and workload faced [36]. Qualitative in-depth interviews, and a rapid assessment design of a quantitative cross-sectional survey were used to answer the two objectives by sampling 14 health facilities from four provinces [36].

**Key findings.**  The studies found a shortage of health workers that provided COVID-19 services in Indonesia [36] (Table 5).

**Table 5.  WISN ratios of the various cadres included in Indonesia.**

| Cadre | WISN ratio |
|---|---|
| Nurse | 0·84 |
| Midwife | 1·31 |
| General practitioner | 0·75 |
| Lab technician | 0·91 |
| Lung specialist | 0·84 |

**Primary lessons and directions forward.** The use of WISN highlighted the need to understand the current scenario and plan future health workforce needs. Based on shortages and workload pressures, the study recommended additional general practitioners, nurses, laboratory personnel, and pulmonary specialists in health facilities that were COVID-19 referral sites [36]. It further proposed for health workers to be transferred from other units or temporary workers to be recruited to boost the number of health workers needed in a pandemic situation [36]. This process also allowed for the realisation that clear workload standards are needed for each hospital-based job and that data standardisation is needed for health workers in all health facilities [36]. As the study results showed differences in service data in each health facility, it recommended for the calculation of workloads and the documentation of a workforce recruitment plan annually [36]. The Indonesian study was limited by the fact that it was undertaken during the pandemic where there were added time constraints; workload standards and detailed job descriptions were not readily available [36].

### Nepal

**Overview and initial implementation.** With the guidance of WHO, Nepal's WISN study assessed prevailing staffing needs for five cadres (medical officers, health assistants, staff nurses, auxiliary nurse midwives and auxiliary health workers) in selected primary care facilities between July 2019 and December 2020 [41]. This was achieved through the determination of workload, quantification of the health workforce needs based on the workload and, the identification of shortages and surpluses of the selected cadres in those facilities [41].

An exploratory sequential mixed-methods research design of qualitative and quantitative research methods was undertaken to answer the objectives [41]. Qualitative research methods used key informant interviews of national, provincial, district and facility managers in selected health facilities to explore staff workload, service standards, scopes of practice, package of services and policy guidelines [41]. The quantitative research method component was a cross-sectional survey that was complemented by a record review of comparisons between the existing and required number of staff (staffing need) [41]. The Health Management Information System (HMIS) records and service registers were analysed to capture data from each health facility [41]. The WISN tool was piloted in nine health facilities located in three (3/7) provinces [41]. The task followed nine (9) sequential steps through collaborations between WHO and the country's Steering committee, Technical Working Group and Expert Working Groups (Fig 2) [41].

**Key findings.** Results were variable between cadres and health facilities [41]. Four Health posts recorded staffing shortages amongst the health assistants with one facility's Health post having none [41]. Equally, very high workload based on the criteria for analysing workload pressure was recorded for health assistants thus resulting in shortages [41]. Extremely low workload pressures were experienced by auxiliary nurse midwives at the health posts indicating staff underutilisation [41]. Overall, staff nurses and auxiliary health workers were adequately staffed (Table 6) [41].

**Primary lessons and directions forward.** Nepal's WISN study had several limitations including a) Difficulty in getting the annual facility health statistics from most of the health facilities visited [41]. This could have led to overestimation or underestimation of staffing requirements [41]. The data collection tool developed by the Expert Working Group and used for data collection in the health facilities only captured some workload and not all [41]. There were significant variations in the activity standards and workload components set by the expert working groups, especially the support and additional activities in the different health facilities [41]. Some facilities did not have documentation of staff availability and absences which are critical to the first step of the WISN methodology [41]. There were errors in the data collected across the facilities and health cadres [41]. The study occurred during COVID-19 lockdowns which impacted data collection [41]. All these limitations had an impact on the accuracy of the results.

Short and long-term recommendations were developed based on the WISN results [41]. Short-term recommendations are focused on administrative and operational actions that could be immediately undertaken by the policymakers [41]. The long-term recommendations may require significant policy, financial and wider stakeholder consultations and involvement before implementation [41].

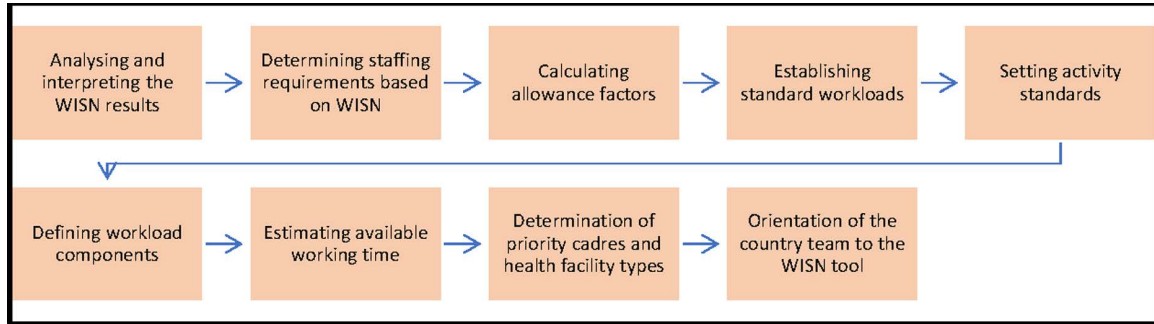

**Fig 2. Nepal's process flow on the implementation of WISN.**

**Table 6. Health workforce staffing needs for various cadres in Nepal.**

| Cadre | Existing staff | Calculated requirements | WISN Ratio | WISN Difference |
|---|---|---|---|---|
| Staff nurses | 7 | 7 | 1 | 0 |
| Auxiliary Nurse Midwives | 34 | 17 | 2 | +17 |
| Auxiliary Health Workers | 20 | 20 | 1 | 0 |
| Medical Officers | 11 | 10 | 1·1 | +1 |
| Health Assistants | 11 | 18 | 0·6 | -7 |

Short-term recommendations include the need to strengthen health facility management; the need to involve more multi-sectoral stakeholders in WISN activities; the development of HRH norms and standards for primary care and the redistribution/transfer of surplus staff in health facilities [41]. Long-term recommendations included the need to align job descriptions with scope of practice of cadres; standardisation of packages of care in primary care facilities; and improved information systems and processes for primary care services [41].

**Sri Lanka**

**Overview and Initial implementation.** Sri Lanka's WISN experiences are based on an interview with a stakeholder, two technical reports and a peer reviewed article [23,37,40]. In Sri Lanka, WISN was planned and implemented under the backdrop of Primary Health Care (PHC) reforms, especially focusing on the health service delivery model [23,37,40]. All the provinces were included in the planning phase [23,37,40]. Sri Lanka uses facility-based staffing standards to calculate the health workforce [23,37,40]. In order to take the actual workload into consideration, they chose to use WISN to calculate workforce needs [23,37,40]. PHC is divided into preventive and curative services which are not co-located [23,37,40]. The central ministry is responsible for the recruitment of health staff, while provincial ministries are responsible to recruit minor staff categories with the concurrence of the central ministry [23,37,40]. The country is implementing health system reforms to strengthen its PHC supported by two projects: a) PHC system strengthening project and, b) Health system enhancement project from 2019.

The Ministry of Health carried out WISN to determine the Public Health Midwife (PHM) cadre requirements for providing UHC to communities in Sri Lanka, and to develop cadre norms for PHC curative services [37].

The study was governed by the national Steering Committee along with the national and provincial Technical Groups [23,37]. Training was conducted to train the provincial and district level medical officers and nursing officers to collect data on activity standards [23,37]. A time and motion study was also conducted to understand the time required for various activities [23,37]. Each province had a training program to understand the details of the data collection process [23,37].

Data from each province were collated at the central level at which activity standards were analysed [23,37]. Detailed terms of reference were documented to guide the technical groups and provincial leaders to implement WISN in their provinces [23,37].

**Key findings.** Country wide WISN ratio for PHMs was 0·58 [23,37]. Based on the initial study, the country required 9846 PHMs, which translated to an additional need of 4130 PHMs in order to provide universal health coverage [37].

The gradual expansion of PHM's scope of work has resulted in accumulated staff gaps [23,37]. Out of the 25 districts, three health districts had a sufficient number of PHMs, while 22 have staff gaps ranging from 0·34 in Ampara to 0·76 in Monaragala (Fig 3) [37].

**Primary lessons and directions forward.** Even though it was acknowledged that consideration needed to be biased towards population health needs and plan health workforce based on a life-course approach for services required within a context, the WISN method was found to be an effective approach [23,37]. In Sri Lanka, WISN findings highlighted shortages and inequities in the distribution of PHMs [23,37]. The country required additional 4130 midwives to provide quality universal reproductive, maternal and child health care services [37]. Further acknowledged is the fact that planning and operationalisation need to work together to build efficiencies and improve health outcomes.

Reflections from key informants of four of the countries where stakeholders were interviewed are summarised in Table 7.

## Discussion

Our study established that all countries except Bangladesh found it difficult to implement WISN with the main challenge being the lack of data on workload standards and the lack of standardised data for activities conducted by staff. All the

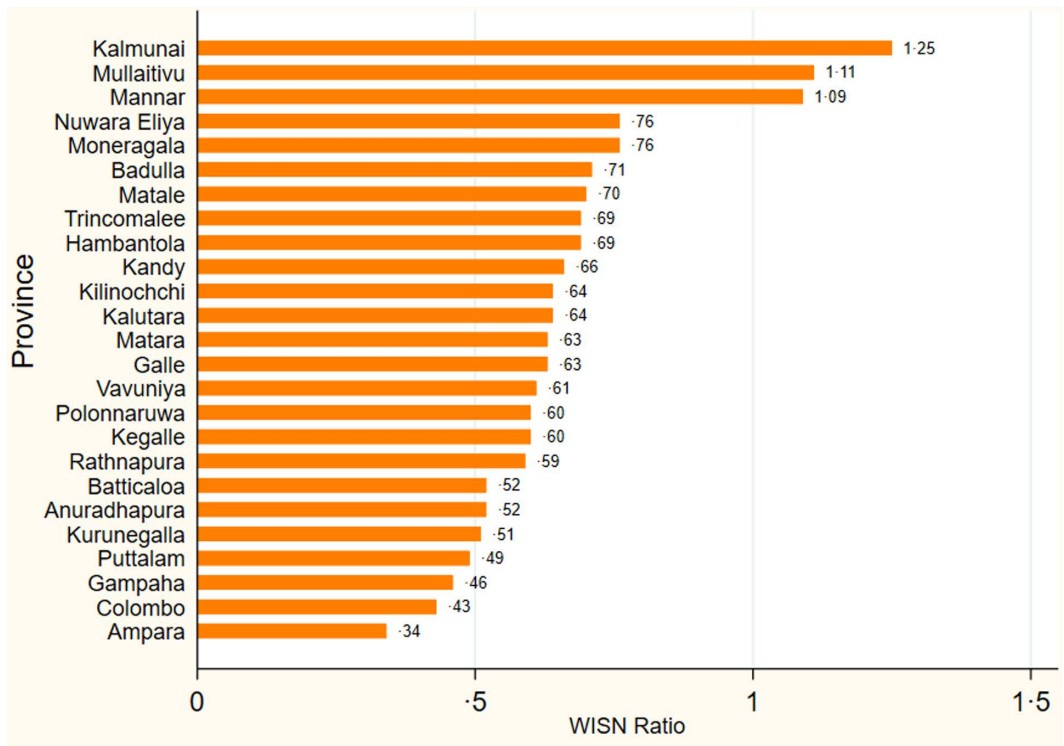

**Fig3. Sri Lanka's WISN ratio for Public Health Midwives (PHMs).**

**Table 7. Reflections from key informants of four of the countries.**

| Bangladesh | Bhutan | India | Sri Lanka |
|---|---|---|---|
| ○ First country in the region to explore WISN (since 2013).<br>○ WISN conducted three times<br>○ Initially, technical expertise provided by WHO Geneva.<br>○ Later, technical expertise have been sourced from USAID, WHO Geneva and the country office.<br>○ The country has sufficient in-house expertise to conduct WISN.<br>○ The ministry set up informant interviews and focus group discussions which helped to collect the primary data. Once the primary data were available, they could refine or tailor them to different settings by discussing service standards in each hospital or health posts.<br>○ A community of practice developed over time has helped people appreciate and understand WISN.<br>○ WISN has been extended to other Government agencies (e.g., the health division of the Ministry of Defence plans to use WISN to develop their workforce needs.<br>○ Setting service standards and activity times is difficult as these vary by context and experience of the healthcare worker.<br>○ WISN needs to be used by health centres in collaboration with the Ministry. | ○ The data on HRH or service standards were not available and this was a challenge in implementing WISN.<br>○ Almost all data were paper-based and there was no record of the time consumption on different activities.<br>○ Paper-based records were time-consuming and incomplete.<br>○ Incomplete data might have led to wrong results. For example, WISN showed an over-supply of nurses, however, the reality is that there was a serious shortage of nurses in the country.<br>○ It was challenging to identify the time taken for activities which were not listed as part of the job description (e.g., administrative responsibilities not captured for community health workers).<br>○ Under-utilised health services are deemed to be in excess by WISN, e.g., dental services in primary care.<br>○ Task-sharing is not captured by WISN as the data is not recorded.<br>○ WISN has emphasised the importance of health information system. | ○ Despite having trained and knowledgeable staff, India still needed technical expertise from WHO to use WISN.<br>○ Having a bottom-up approach would make the findings of WISN acceptable and improve health service delivery and improve health outcomes.<br>○ India needs to establish service standards and a Hospital Management Information System which will provide specific information required for WISN assessment.<br>○ WISN is not recommended for states or facilities with known workforce shortages. For such facilities, the first step is to focus on and achieve IPHS. Once a facility has sufficient numbers, then WISN can be used to fine tune the required number of health workers. | ○ While cadre norms for PHC facilities were calculated and presented in a report, WISN ratios were not calculated for all the primary care workforce cadres (except for midwives).<br>○ The report included the workload components and the activity standards.<br>○ WISN ratios were not included in the reports<br>○ A strong trade union presence helped increase endorsement of final report.<br>○ The WISN results may not be implemented to strengthen PHC workforce due to the separation of authorities between those who planned for it and those responsible for PHC operationalisation.<br>○ As preventive and curative services are separate, some activities are not coordinated between these two areas of primary care.<br>○ Sri Lanka's PHC reforms are funded by development agencies who financed the PHC infrastructure without focusing on the wider system related issues, including workforce. |

HRH = Human Resources for Health; IPHS = Indian Public Health Standards; PHC = Primary Health Care.

countries except Nepal and for few cadres in Bhutan indicate significant shortages in staffing. Except Bhutan where there were questions about the accuracy of the findings, the results of the study from other countries were accepted by the policymakers. Nonetheless, no country actioned the recommendations or followed up to evaluate if the sanctioned changes were implemented by the health facilities. Our study found that while WISN was developed to provide reliable evidence for workforce planning, it has some limitations in the context of LMICs. This is especially true for PHC settings where it does not consider the motivation of the workforce and the external factors such as availability of health system infrastructure and resources [18], availability of facilities in rural regions needed to increase retention. Furthermore, WISN does not consider the variations in the demand for services, task-sharing between team members and motivation to work in underserved areas [9,18,43]. The current use of WISN also does not consider variations in skill set, competencies and performance of the workforce.

WISN needs to be planned and implemented using a systems lens taking into consideration the macro (national health policy), meso (health system infrastructure) and microenvironment (team dynamics, interpersonal relationships). Defining and setting service activities and standards are critical steps in the WISN methods as they directly impact the estimation of staffing requirements [43]. Most countries conduct either a Delphi method or a series of discussions to develop activity standards. This was the most challenging step for all countries and each of them undertook detailed planning workshops, group discussion and also conducted time and motion studies. Care needs to be taken to include experts from various

facilities and competencies as the final activity standard is based on the consensus of these discussions which may impact under of over estimation of staffing requirements for health facilities [9]. Service activities and standards depend on the available infrastructure, working conditions, health service packages, team skill mix and scope of practice, and varies widely across different contexts within and across countries [43]. Policymakers and stakeholders have acknowledged this challenge [6,9,17,18,44].

A recommendation for the implementation of WISN is to advocate its use among the health workforce so that they are convinced about the usefulness of WISN. Our study found that while data collection was planned well with technical expertise from WHO, it followed a top-down approach with little enthusiasm from the health workforce. Lack of stakeholder consultation and involvement often leads to resentment among health workers [7]. The use of WISN needs to be institutionalised so that the health workforce is involved from the beginning, making the process a 'bottom-up' approach with technical support from experts and logistical support from the leadership. This may take time and may involve multiple rounds of deliberations with the key stakeholders. Without adequate stakeholder consultation and creating an environment of trust which comes from the assurance of the leadership at higher levels, followed by implementation of the recommendations, the health workforce may not be convinced that WISN will improve anything. A WISN study conducted in Uganda highlighted that involving key stakeholders from the start increased ownership and acceptability of the results [18]. Studies have shown that during the early years of implementing WISN, countries find it resource and time intensive [11,17,43], however, there are opportunities to streamline the process. Countries can be innovative in developing their service and activity standards by involving their health workforce from the beginning and using digital tools to capture activity and service standards and conduct time and motion studies [45].

WISN was developed as a human resource management tool to allow countries to have the right number of people, with the right skills, in the right place and at the right time. This knowledge would help the health workforce as policymakers would have a better understanding of their workload and make provisions and budgetary recommendations for adequate numbers [25]. This would also be beneficial to the communities who would have easy access to trained health workers who have sufficient time to devote towards their patients, thereby helping countries to achieve universal health coverage.

## Strengths and limitations

WISN could be successfully used to understand staffing needs in five out of the six countries under study in the SEAR region. We reviewed WISN reports and interviewed five policymakers from four countries (Bangladesh, Bhutan, India and Sri Lanka). Having insights from decision makers helped understand the facilitators and barriers in the use of WISN and reflections about the future use of WISN and how it can be used to strengthen the country's health workforce database. One limitation of our study is that we could not interview key stakeholders from all countries. Even though the reports and literature for Indonesia and Nepal were comprehensive, they could not completely compensate for the lack of interviews. Furthermore, despite attempts to ensure comparability of findings and recommendations between countries, this was not always the case. However, this study was able to achieve its objective of ascertaining whether the recommendations were utilised beyond the WISN determinations.

Using WISN has helped all the stakeholders acknowledge the need for documentation, to define workload components, set activity standards, and standard workload. None of the countries had costed the implementation of WISN. Furthermore, due to discrepant data between a 2020 article [23] and the 2021 report on PHMs in Sri Lanka [37], this research used the report.

## Recommendations

Many low- and middle-income countries use facility-based staffing standards rather than workforce indicator-based needs. Our study demonstrates that the key reason for using facility-based staffing is the lack of documentation regarding activity and workload standards in all the five countries. Therefore, our recommendations are:

○ Resource-poor countries need to have national standards for facility-based staffing needs (e.g., Indian Public Health Standards). Countries with a critical shortage of the health workforce should first prioritise addressing those shortages. WISN should be recommended if facilities have sufficient staff at each level.

○ Strengthen health workforce data by digitising health workforce records and strengthening the workforce database.

○ Document workload and activity standards using a range of experts from diverse contexts (competencies and different facility levels) in countries.

○ Document tasks shared between team members, especially in primary health care.

○ Conduct time and motion studies across different contexts in countries to capture the work done by each member (administrative and clinical tasks) and by teams.

○ WISN should be internalised by the health workforce, and the staff should be trained to do it themselves with some external checks.

○ Include all the cadres within each facility for WISN analysis.

○ Develop in-house capacity to use WISN and initiate communities of practice within and across countries to exchange knowledge about the use of WISN.

○ WISN should be carried out only if Governments have the capacity to use the recommendations by allocating sufficient funds for additional staff.

○ The above recommendation will most likely only be possible if there is cooperation between various government ministries involved on employment of staff in the public health sector. Such ministries include the Ministry of Health at national and sub-national level, Treasury, and the ministry that is responsible for employment in the public service, e.g., Ministry of Public Service administration, Ministry of Labour, etc. All these stakeholders should be involved from early on and should then review the recommendations together to discuss strategies on how the recommendations will be phased in over time and operationalised.

## Conclusions

WISN is a useful tool, however, countries need to first strengthen their health workforce information systems and digitise workload standards. Once these data are digitised, WISN would be a powerful tool to help policymakers plan, train, recruit and equitably distribute health workers to attain universal health coverage. Furthermore, the government treasury and the employing ministry must be involved at early stages of the planning process to ensure easy phasing in and adaptation of recommendations.

### Research in context

### Evidence before this study

South Asia has some of the most acute health workforce shortages in the world. The World Health Organization (WHO) developed the Workload Indicators of Staffing Need (WISN), a method to calculate human resources needs for health systems, considering the healthcare delivery models and population demands. It provides two indicators to assess staffing needs: (a) the gap between current and required number of staff and (b) a measure of workload pressure on health workers. Staffing requirements are calculated by the workload at the facility and the standard workload for a cadre.

## Added value of this study

This is the first comprehensive multi-method study that assesses implementation of WISN in more than four South East Asian countries. Whilst all countries were able to determine the human resource deficit and/or surplus, most have not been able to implement the recommendations. Service activity standards and data were not readily available for making WISN determination.

## Implications of all the available evidence

Countries need to first strengthen their health workforce information systems and digitise workload standards before WISN can be used effectively.

## Supporting information

**S1 Text. Interview guide.**
(PDF)

## Acknowledgments

We express our gratitude for the support received from the World Health Organization South East Asia Regional Office, and governments of the studied countries. This work would not have been possible without assistance from Ibadat Dhillon, Thinley Zangmo, Tashi Tobgay and Md Nuruzzaman.

## Author contributions

**Conceptualization:** Sikhumbuzo A. Mabunda, Rohina Joshi.

**Data curation:** Sikhumbuzo A. Mabunda, Rohina Joshi.

**Formal analysis:** Sikhumbuzo A. Mabunda, Rohina Joshi.

**Funding acquisition:** Sikhumbuzo A. Mabunda, Rohina Joshi.

**Investigation:** Sikhumbuzo A. Mabunda, Rohina Joshi.

**Methodology:** Rohina Joshi.

**Project administration:** Sikhumbuzo A. Mabunda, Rohina Joshi.

**Resources:** Rohina Joshi.

**Software:** Sikhumbuzo A. Mabunda, Rohina Joshi.

**Supervision:** Rohina Joshi.

**Validation:** Rohina Joshi.

**Visualization:** Sikhumbuzo A. Mabunda.

**Writing – original draft:** Sikhumbuzo A. Mabunda.

**Writing – review & editing:** Sikhumbuzo A. Mabunda, Mona Gupta, Roshan Sampath, Rohina Joshi.

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
