## [Decision Letter · Decision Letter 0]

PGPH-D-24-01831

The implementation of the Workforce Indicators of Staffing Needs (WISN) method to improve access to health workforce in selected South-East Asian countries

Dear Dr. Mabunda,

Thank you for submitting your manuscript to PLOS Global Public Health. After careful consideration, we feel that it has merit but does not fully meet PLOS Global Public Health’s publication criteria as it currently stands. Therefore, we invite you to submit a revised version of the manuscript that addresses the points raised during the review process.

We look forward to receiving your revised manuscript.

Kind regards,

Jianhong Zhou

Staff Editor

Journal Requirements:

Additional Editor Comments (if provided):

Reviewers' comments:

Reviewer's Responses to Questions

**Comments to the Author**

1. Does this manuscript meet PLOS Global Public Health’s publication criteria ? Is the manuscript technically sound, and do the data support the conclusions? The manuscript must describe methodologically and ethically rigorous research with conclusions that are appropriately drawn based on the data presented.

Reviewer #1: Yes

Reviewer #2: Yes

2. Has the statistical analysis been performed appropriately and rigorously?

Reviewer #1: N/A

Reviewer #2: N/A

3. Have the authors made all data underlying the findings in their manuscript fully available (please refer to the Data Availability Statement at the start of the manuscript PDF file)?

Reviewer #1: Yes

Reviewer #2: Yes

4. Is the manuscript presented in an intelligible fashion and written in standard English?

Reviewer #1: Yes

Reviewer #2: Yes

5. Review Comments to the Author

Reviewer #1: Dear authors,

Congratulations for a well written article. You have done a very good job!

I have very few recommendations as you done a great job:

- in the introduction, you focus quite a lot on WISN, but do not provide very much context on the countries studied, which would allow the reader to better understand your results;

- it would be beneficial to present what was the reasoning for the choice of methodology.

Thank you!

Reviewer #2: Abstract

Although there are no requirements for separate subheadings, it will be useful to keep the balance. Please indicate the need for this research in one of the first sentences (before the methods) – it is stated “The Workload Indicators of Staffing Needs (WISN) is a tool used to estimate health workforce requirements for national or sub-national regions and health facilities. We determined the implementation of WISN for six countries in the World Health Organization South-East Asia Region (WHO-SEAR).” However, it is not clear why the research is crucial, its p[possible impact and/or the current gaps are.

“We used a multi-method approach comprising qualitative, in-depth, semi-structured interviews, literature reviews and report or document reviews” – what do authors mean by “and report or document reviews”? Report analysis and document reviews?

Additionally, the authors present information mainly through interviews with stakeholders (in-depth or semi-structured?) but no information on how various literature reviews were conducted.

On the other hand, “Transcription errors were corrected by the first author by listening to the audio and editing all identified Errors” is too much information and is better placed in the results section of the manuscript.

Keywords:

“WISN; Health workforce OR Human resources for health OR HRH AND Planning; Health systems” – the Boolean operators are redundant here. Please delete.

Introduction:

“Considering the need for efficient and effective health workforce management, the World Health Organization (WHO) developed the Workload Indicators of Staffing Needs (WISN), a method to calculate human resource requirements for health systems, considering the healthcare delivery models and population demands [5].” – needs restructuring, it is challenging to follow the logic here.

Although it is generally clear that this study is needed, it is worth elaborating on particular (health) related problems in South Asia. Why is there acute health workforce shortage and what does it mean for South Asia (perhaps elaboration on the most prominent diseases, public health issues, etc).

Have there been any attempts to clarify staffing needs in the past? (possibly comparable to Workload Indicators of Staffing Needs (WISN) assessment).

What is the aim/objective of this research? Are there any research questions the authors want to address?

Materials and Methods

The authors are suggested to re-write the methods section and add relevant references for the data collection methods used in this study (multi-methods approach). The section needs extensive revision.

Why/when semi-structured and/or in-depth interviews? Regarding the literature and report reviews, are there any publication date limits? Language? Why this review methods were preferred?

Also the authors reveal some results in the methods section “Five interviews from four countries were conducted virtually by RJ using Microsoft Teams and lasted an average of 47 minutes. The research team did not manage to get into contact with an Indonesian and Nepalese focal people for WISN using multiple strategies. Two participants were interviewed separately for Bhutan” – please keep the methods consistent.

“Data were audio-recorded and transcribed using Microsoft Teams after obtaining consent from participant” – this information appeared earlier in the manuscript.

“The last Table in the Results section summarises reflections on the process and the utility of

WISN by the five interviewed participants from four of the countries” – What is the number of the table you are referring to?

Sampling and, Identification of Literature and Reports – it is unclear if there were any inclusion or exclusion criteria. Although sampling bias are not avoidable, please reveal essential characteristics of involved stakeholders.

“Medline and CINAHL searches were conducted to identify all literature on WISN from these five WHO South East Asian countries” – dates/limits? Why did the authors use only scientific databases? Especially when there is a statement of applying report reviews?

“Role of the funding source This work was funded by WHO SEARO. Data collection, analysis and interpretation was led by researchers in consultation with country-based researchers and policymakers.” – it seems not appropriate in the methods section of the manuscript

Resuls.

It is not completely clear how the multi-methods approach was triangulated. How did the authors compensate for the lack of data from the stakeholder interviews in Indonesia and Nepal?

Additionally, there is some inconsistency in data presentation across the countries: for example, there is an overview (table 4) Short, medium and long-term WISN recommendations in Bhutan, this is the only country that provides this.

Overall, there is inconsistency in the presentation of all six countries, which makes it difficult for the reader to understand why the analysis are not comparable. The authors claim to use semi-structured and in-depth interviews using the interview guide. Additionally the review analysis could support the consistency of data presentation (thems).

Bangladesh: Workload indicator staffing needs ratio and health workforce requirements for Bangladesh.

Bhutan: Short, medium and long-term WISN recommendations in Bhutan.

India: WISN Ratio results for four cadres in two Indian health facilities

Indonesia: WISN ratios of the various cadres included in Indonesia

Nepal: Nepal's process flow on the implementation of WISN; Health workforce staffing needs for various cadres in Nepal; Short and long-term recommendations (without a table like in Bhutan).

Sri Lanka: Sri Lanka’s WISN ratio for Public Health Midwives (PHMs)

“Reflections from four of the countries are summarised in Table 7” – it is worth mentioning that the opinions from the stakeholders are based on one interview per country (!) (with an exception of 2 in Bhutan), so the authors should be careful with the statement “Reflections from four of the countries” – instead those are the single opinions.

General for the results: although the majority of the findings are exciting and informative, the authors should restructure the section, especially for single countries. The tables and figures are diverse, covering various aspects. For example, for Sri Lanka there is a table for WISN ratio for Public Health Midwives, it is particular profession. Also, it is not clear whether PH is considered to be a part of health workforce in the country. If so, please elaborate on the relevance of PH workforce (and its definition) in the introduction.

Generally, it is challenging to go through separate county information; please help the reader to structure the information (to enhance the generalizability of the findings).

Discussion:

I believe that WISN could be successfully used to understand staffing needs. However, it is essential to understand the limitations of this study's design. Only four out of six countries provide insights from single interviews, making eliminating subjective opinions extremely challenging.

Also, there could be an interplay and definition clash between the health workforce and the PH workforce (e.g., PH midwives).

The authors present recommendations (general for all 6 countries?); how does the reader deal with the recommendations provided by representatives from Bhutan and Nepal? Could you please clarify the generalizability issues (transferability of the results)?

General comments:

There is confusion about the use of Level 1-2-3 heading throughout the manuscript (e.g. the results are the same level heading as subsections within it)

6. PLOS authors have the option to publish the peer review history of their article (what does this mean? ). If published, this will include your full peer review and any attached files.

**Do you want your identity to be public for this peer review?** For information about this choice, including consent withdrawal, please see our Privacy Policy .

Reviewer #1: No

Reviewer #2: No

---

## [Decision Letter · Decision Letter 1]

The implementation of the Workforce Indicators of Staffing Needs (WISN) method to improve access to health workforce in selected South-East Asian countries

PGPH-D-24-01831R1

Dear Dr. Mabunda,

We are pleased to inform you that your manuscript 'The implementation of the Workforce Indicators of Staffing Needs (WISN) method to improve access to health workforce in selected South-East Asian countries' has been provisionally accepted for publication in PLOS Global Public Health.

Best regards,

Julia Robinson

Executive Editor

Reviewer Comments (if any, and for reference):

Reviewer's Responses to Questions

**Comments to the Author**

1. If the authors have adequately addressed your comments raised in a previous round of review and you feel that this manuscript is now acceptable for publication, you may indicate that here to bypass the “Comments to the Author” section, enter your conflict of interest statement in the “Confidential to Editor” section, and submit your "Accept" recommendation.

Reviewer #2: All comments have been addressed

2. Does this manuscript meet PLOS Global Public Health’s publication criteria ? Is the manuscript technically sound, and do the data support the conclusions? The manuscript must describe methodologically and ethically rigorous research with conclusions that are appropriately drawn based on the data presented.

Reviewer #2: Yes

3. Has the statistical analysis been performed appropriately and rigorously?

Reviewer #2: N/A

4. Have the authors made all data underlying the findings in their manuscript fully available (please refer to the Data Availability Statement at the start of the manuscript PDF file)?

Reviewer #2: Yes

5. Is the manuscript presented in an intelligible fashion and written in standard English?

Reviewer #2: Yes

6. Review Comments to the Author

Reviewer #2: Dear Athors,

Thank you for sdressing the comments and congrartulations on your work.

7. PLOS authors have the option to publish the peer review history of their article (what does this mean? ). If published, this will include your full peer review and any attached files.

**Do you want your identity to be public for this peer review?** For information about this choice, including consent withdrawal, please see our Privacy Policy .

Reviewer #2: No
